# Rapid accelerations of Antarctic Peninsula outlet glaciers driven by surface melt

Peter A. Tuckett[1], Jeremy C. Ely [1]*, Andrew J. Sole [1], Stephen J. Livingstone [1], Benjamin J. Davison[2], J. Melchior van Wessem[3] & Joshua Howard[1]

Atmospheric warming is increasing surface melting across the Antarctic Peninsula, with unknown impacts upon glacier dynamics at the ice-bed interface. Using high-resolution satellite-derived ice velocity data, optical satellite imagery and regional climate modelling, we show that drainage of surface meltwater to the bed of outlet glaciers on the Antarctic Peninsula occurs and triggers rapid ice flow accelerations (up to 100% greater than the annual mean). This provides a mechanism for this sector of the Antarctic Ice Sheet to respond rapidly to atmospheric warming. We infer that delivery of water to the bed transiently increases basal water pressure, enhancing basal motion, but efficient evacuation subsequently reduces water pressure causing ice deceleration. Currently, melt events are sporadic, so efficient subglacial drainage cannot be maintained, resulting in multiple short-lived (<6 day) ice flow perturbations. Future increases in meltwater could induce a shift to a glacier dynamic regime characterised by seasonal-scale hydrologically-driven ice flow variations.

[1] Department of Geography, The University of Sheffield, Sheffield, UK. [2] School of Geography and Sustainable Development, University of St Andrews, Andrews, UK. [3] Institute for Marine and Atmospheric research Utrecht, Utrecht University, Utrecht, The Netherlands. *email: j.ely@sheffield.ac.uk

The melting of snow and ice creates networks of meltwater streams and ponds on glacier and ice-sheet surfaces[1–3]. Surface meltwater is known to influence the dynamics of many glaciers[4–7] and portions of the Greenland Ice Sheet[8,9]. At these locations, drainage of surface water to the ice-bed interface impacts basal water pressure, leading to variations in ice motion on sub-daily to decadal timescales[1,4,7–13]. This mechanism couples ice flow to atmospheric processes over a range of timescales down to the sub-diurnal level, making affected ice masses respond rapidly to changes in atmospheric circulation patterns induced by climate change[9].

Surface meltwater has recently been reported to be widespread across the Antarctic Ice Sheet[3]. Antarctic-wide melting is projected to double by 2050, with the greatest increases in meltwater production expected across the Antarctic Peninsula, where atmospheric warming is already increasing surface melting[14,15]. Surface meltwater had catastrophic consequences for some ice shelves, where meltwater expedites ice-shelf disintegration by hydraulically driven fracture[16]. However, whether surface meltwater impacts the behaviour of grounded ice in Antarctica, implying a rapid and direct coupling of ice dynamics with atmospheric conditions, is currently unknown[17].

In this paper, we use high-resolution satellite-derived ice velocity data, optical satellite imagery and regional climate modelling to study five Antarctic Peninsula outlet glaciers and ascertain whether surface meltwater drains to the bed of these glaciers, influencing dynamics at the ice-bed interface. The data reveal that multiple short-lived (<6 day) rapid and large accelerations (in some cases 100% greater than mean annual ice velocity) termed speed-up events occur. That the relative magnitude of speed-up events increases away from the marine margin, and that tidal fluctuations, iceberg calving events and sea-ice break-up are asynchronous with speed-ups, leads us to rule-out marine processes as a cause. Instead, the spatial pattern of speed-ups, their concurrence with periods of modelled surface melting, and observations of potential routes for surface meltwater to reach the bed from optical satellite imagery are consistent with a surface meltwater trigger. For the first time, we show that surface meltwater affects the dynamics of grounded ice in Antarctica, proving a mechanism by which atmospheric and ice-dynamic processes are coupled over short-timescales.

## Results

**Ice velocity.** We used feature tracking of Sentinel-1 radar imagery to generate velocity maps at 6-day intervals for five marine-terminating outlet glaciers on the Antarctic Peninsula (Fig. 1), during the period October 2016 to April 2018 (Methods). Four of the glaciers, Drygalski, Hektoria, Jorum and Crane, are on the eastern side of the Antarctic Peninsula, while Cayley Glacier is on the west side (Fig. 1). For each glacier, velocity was analysed across a series of 1 km square regions of interest (ROIs), extending inland from the glacier's grounding-line. Glacier-averaged ROI velocities reveal three types of velocity fluctuation. First, large-scale seasonal variations, which decrease in magnitude away from the ocean. Second, background velocity fluctuations of ≈50 m a$^{-1}$ magnitude. Finally, short-lived rapid accelerations in ice velocity to values >20% greater than the annual mean (Fig. 2a). The latter of these we refer to here as speed-up events and are the focus of this paper. Despite being separated by over 100 km, all five glaciers experienced near-synchronous speed-up events in March 2017, November 2017 and March 2018 (Fig. 2a); for example, velocities averaged across all ROIs on Drygalski Glacier (Fig. 1c) increased by 300 m a$^{-1}$ (≈23% greater than the annual mean) in November 2017 (Fig. 2a). Larger accelerations were recorded in individual ROIs (Supplementary Figs. 1–4), with

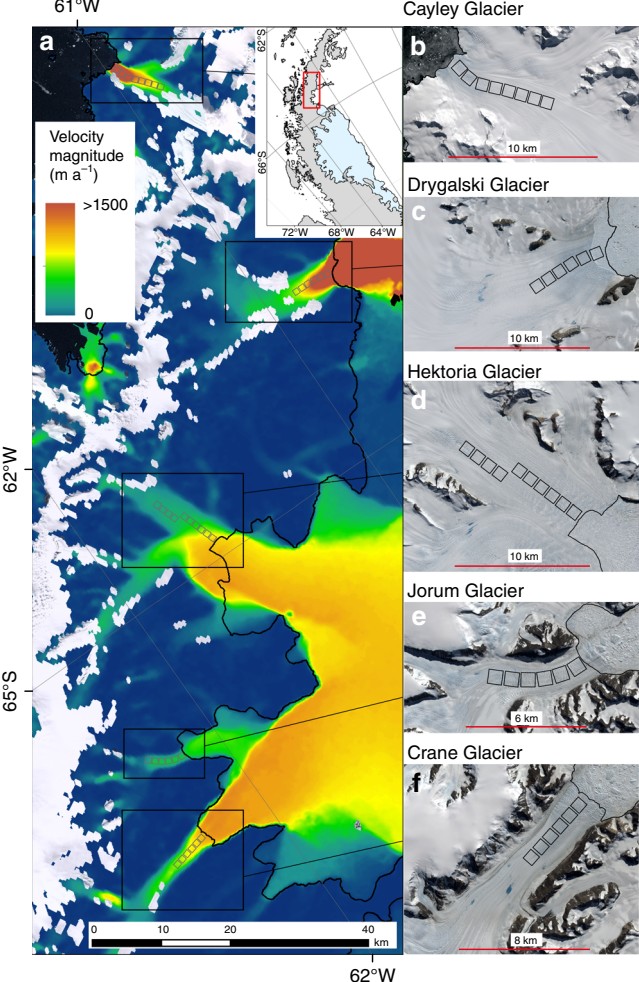

**Fig. 1** Mean ice velocity map and location of studied glaciers. **a** Mean velocity magnitude across the study area, between October 2016 and mid-April 2018. Inset shows the location on the peninsula, and the black line is the grounding line[45]. **b–f** Landsat 8 satellite images of glacier surfaces. The black boxes outline the regions of interest. Note the visible water on the glacier surfaces

a 400 m a$^{-1}$ (100% greater than the annual mean) speed-up occurring 9 km from the grounding line of Hektoria Glacier during the November 2017 event (Supplementary Fig. 4c). These data indicate that speed-up events were short-lived, typically lasting for 6 days or less. Most of the speed-up events were followed immediately by a slow-down before velocities return to below pre-event values. No significant slow-down events were detected in the absence of a speed-up. Slow-down and speed-up events were similar in duration (≤6 days), but the velocity change associated with the slow-downs was typically smaller (Fig. 2a, c; Supplementary Figs. 1–4). Such speed-up events have not previously been reported from Antarctica.

**Marine processes.** Two sets of processes could be the cause of speed-up events, either marine processes and/or surface meltwater drainage to the base of the glacier. Marine processes, such as tidal fluctuations[18], seasonal sea-ice break-up[19] or iceberg calving events[20], can all lead to changes in ice buttressing forces. If these processes were the trigger for the observed speed-up events (Fig. 2), we would expect an increase in the relative magnitude of speed-up events closer to the glacier terminus and changes in marine conditions to coincide with speed-up events.

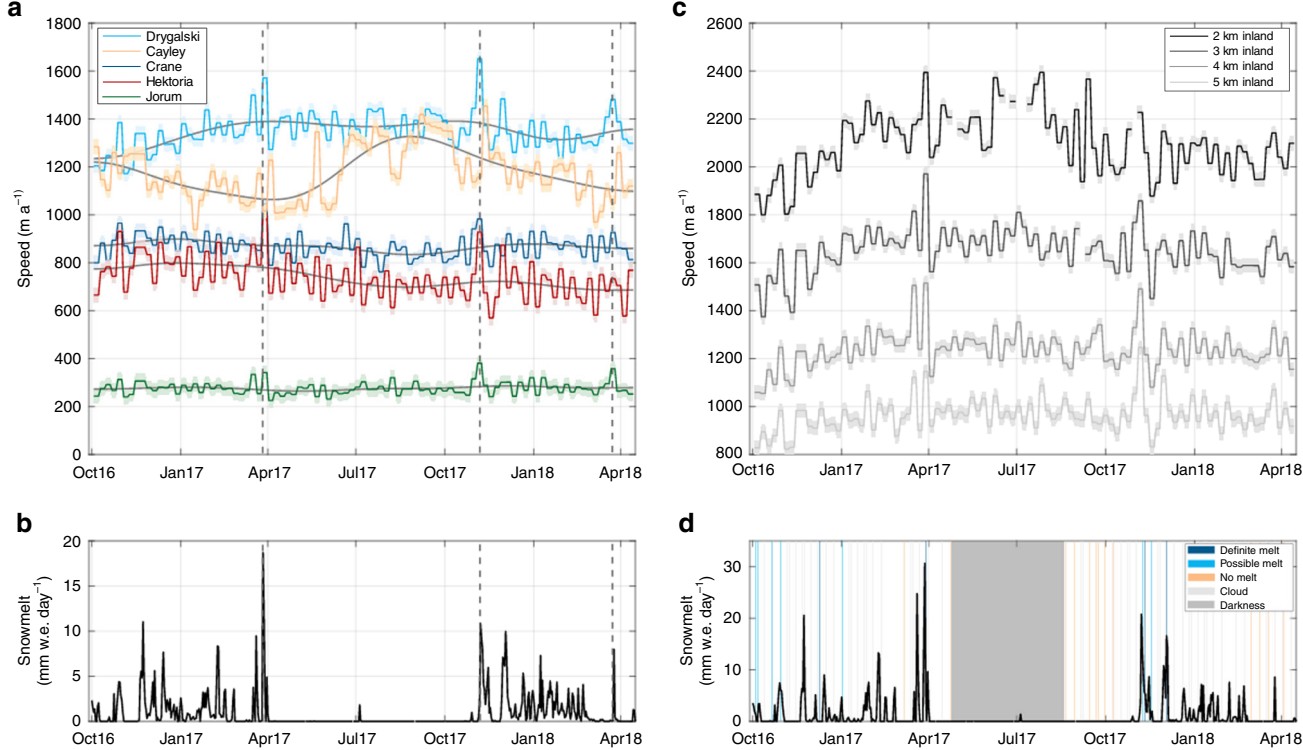

**Fig. 2** Velocity and modelled surface melt of studied glaciers. **a** Velocity variations at multiple glaciers display prominent and coincident speed-ups, defined in the text, indicated by dashed lines, which correspond with spikes in modelled melt **b**. Modelled melt in **b** is averaged across all glaciers. **c** velocity in separate ROIs at Drygalski glacier (Fig. 1). Velocity uncertainties are shown by shaded envelopes around the data. These speed-up events correspond with spikes in modelled melt and observations of surface meltwater across Drygalski glacier in Landsat 8 imagery **d**

At each glacier, many speed-up events became larger compared to the annual mean velocity within ROIs further from the marine margin (Figs. 2c and 3; Supplementary Figs. 1–4). The increase in relative magnitude upglacier was particularly apparent for the November 2017 event (Fig. 3). This is opposite of what would be expected if marine processes were the cause of speed-up events. Analysis of the periodicity of the velocity variations indicates that low magnitude tidal (14-day) and seasonal variations were apparent close to the grounding-line but that they diminished in magnitude inland (Fig. 4). These periodic influences on ice motion contrast with the irregular occurrence of relatively larger glacier speed-up events, which occur beyond the region where a tidal influence on ice motion is apparent (Fig. 4). To further investigate the influence of marine processes on ice velocity, we used satellite imagery to record the calving front position and sea ice/shelf conditions of the studied glaciers (Methods). For all five glaciers, frontal position was remarkably stable during the study period, typically varying by <200 m (Supplementary Figs. 5, 6). The clearest temporal patterns during this time period were the minor advances of Crane and Hektoria glaciers, but no clear pattern between the timing of speed-up events and front position change occurred (Supplementary Figs. 5 and 6). At the southern-most three glaciers (Crane, Jorum and Hektoria), sea ice was present in front of the glaciers for the entirety of the study period, with the sea ice edge in this region remaining over 50 km from the glacier fronts (Supplementary Fig. 7). Given the increase in the relative magnitude of speed-up events away from the marine margin and the lack of synchrony between speed-up events and ice shelf, sea ice or tidal processes, we find it unlikely that any of the investigated marine processes caused the observed velocity fluctuations.

**Regional climate modelling**. To investigate whether there was a temporal correspondence between surface meltwater generation and the observed speed-up events, we compared our velocity data to surface melt rates from a regional climate model[21]. The modelled melt season lasted from October to April, and a small Föhn wind-induced melt event was modelled in July 2017, during the austral winter. Multiple large (>3 mm w.e. day$^{-1}$) but short-lived (<1 week) spikes in melt are modelled, separated by periods of little-to-no melting. The large spikes in melt are coincident with austral summer Föhn events. There is a striking qualitative correspondence between periods of modelled surface melting and speed-up events (Fig. 2), with speed-up events generally occurring during large modelled melt events (e.g., March 2017), or during a less intense melt event preceded by a period of limited or no melt (e.g., November 2017; March 2018). The lowest melt rates were modelled for the western Antarctic Peninsula, where melting is suppressed due to high snowfall rates[21]. Despite this, Cayley Glacier still exhibits several speed-up events coincident with the spikes in modelled surface melting (Supplementary Fig. 3). Speed-up events also occurred during periods when water was visible in satellite imagery of the ice surface (Fig. 2c). The correspondence between large and/or initial melt events and glacier speed-up is consistent with theoretical predictions[22] and observations[1,9] of ice flow variations induced by surface melt drainage to the bed on the Greenland Ice Sheet and Arctic Glaciers[6,23]. Subglacial hydraulic efficiency adapts to accommodate meltwater inputs at timescales longer than the melt events[22]. This means that during periods of rapidly varying meltwater flux, such as large or initial melt events, more water is delivered to the ice-bed interface than can be evacuated by the subglacial system. This leads to a spike in basal water pressure, and an increase in basal sliding, which we suggest

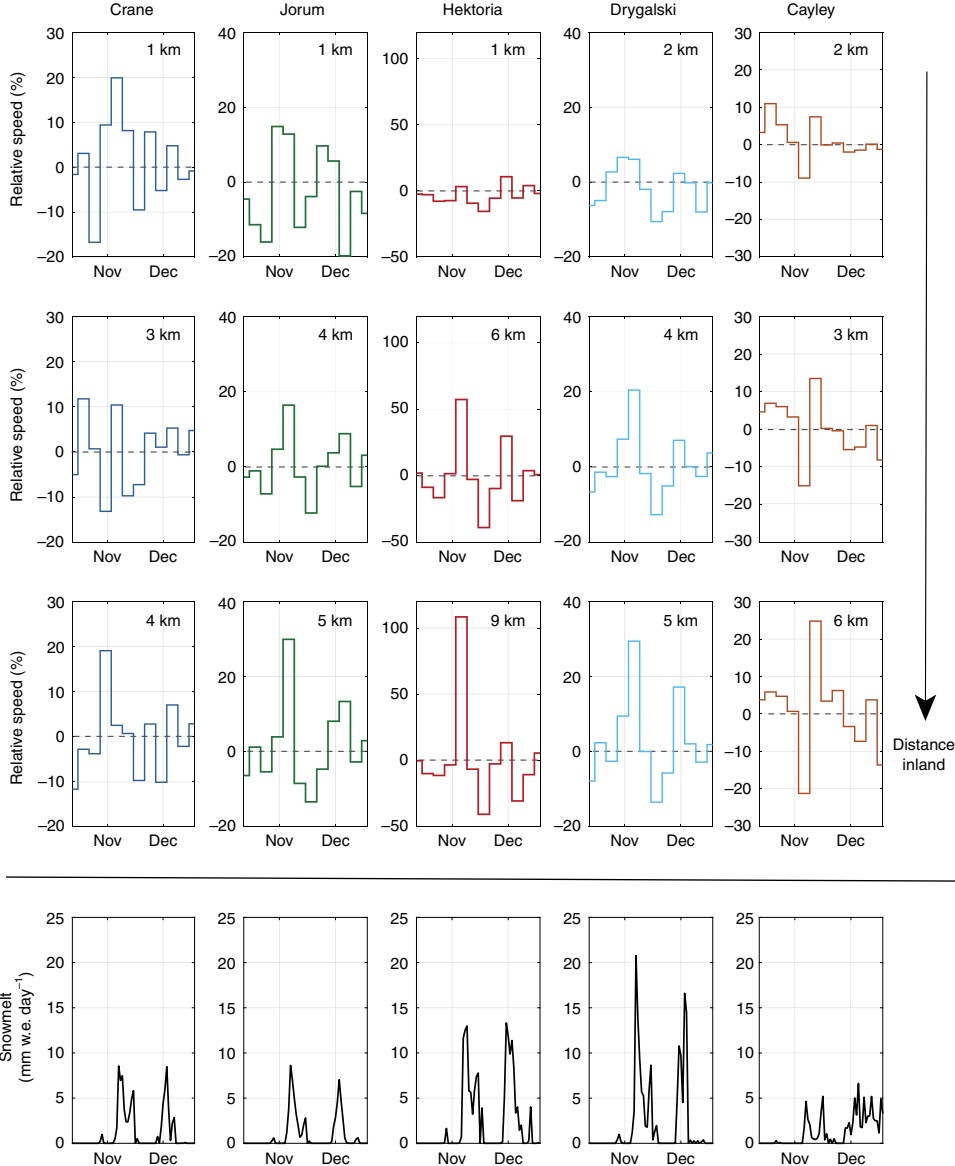

**Fig. 3** Relative velocity of studied glaciers during the November 2017 speed-up event. Note how the relative magnitude of the speed-up increases further away from the marine margin. Ice velocity was detrended using a 144-day running mean. Melt rates are mean values for every ROI studied at the respective glacier

causes the observed speed-up events. We also observe that prolonged periods of high volume, but low variability melt did not elicit a large or extended velocity response (Supplementary Figs 1–4). During these periods, the subglacial system is likely to have had time to adapt to steady surface meltwater inputs.

**Optical satellite imagery**. To help determine whether surface melt reaches the bed of the studied glaciers, we examined the optical satellite image record (Methods). Surface melt features were observed upglacier of the grounding line of each glacier, including small lakes fed by streams and water-filled crevasses (Fig. 5; Supplementary Figs. 8–12). Although lake refreezing was observed to occur in some localities indicating long-term surface storage of water (Fig. 5f), we also recorded multiple occasions where lakes disappeared between satellite images, which we interpret as drainage events. While there is a chance that the lakes could have drained over the ice surface, or into local firn layers, they possessed no apparent outlet stream, coincident waxing and waning of nearby lakes was not observed, and the lake beds

following drainage were heavily crevassed (Fig. 5b, c; Supplementary Figs. 5g, 10c, 11e). In these instances, we suggest that lakes drained into the ice, potentially reaching the bed. Abrupt stream terminations (which may either indicate moulins or points where water drains into firn) and meltwater-filled crevasses are additional potential routes for surface meltwater to access the bed. The pressure exerted by water within crevasses can cause hydrofracture[24], which creates surface-to-bed hydraulic connections.

## Discussion

Given the spatial pattern of speed-up events, their temporal coincidence with modelled melt, lack of coincident variations in glacier frontal position, and observations of surface meltwater drainage, we infer that the observed speed-up events were caused by surface meltwater drainage to the base of the glaciers. The lack of a detectable lag (based on the 6-day temporal resolution of our velocity data) between spikes in snowmelt and speed-up events suggests that the surface is well-connected to the bed, with

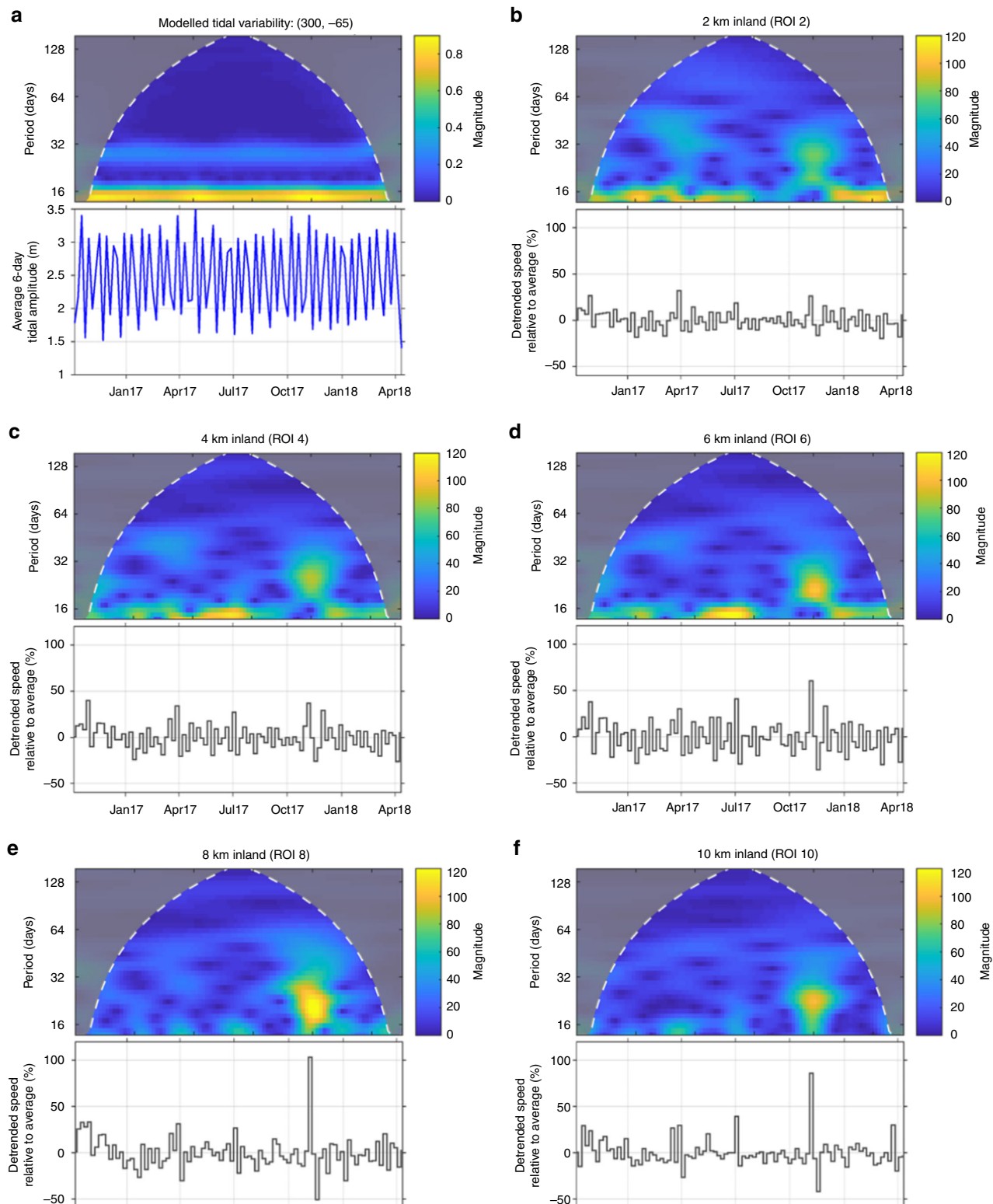

**Fig. 4** Periodicity of tides and velocity data. Scalograms (upper) and time series (lower) of tidal **a** and detrended velocity data **b–f** of Hektoria Glacier. Scalograms show the strength of signal variability (a strong variability is represented by yellow colours) at each period over time. White line delimits the cone of influence, beyond which edge-effects occur. Note the prominent 14-day periodicity of the tidal model **a**. This signal is apparent in the velocity data close to the grounding-line **b**, **c**, but dissipates further inland **e**, **f**, where a more sporadic signal dominates, corresponding to the speed-up events we observe **e**, **f**. Note how the relative magnitude of the meltwater-induced speed-up event increases away from the sea up to 8 km before starting to decrease again

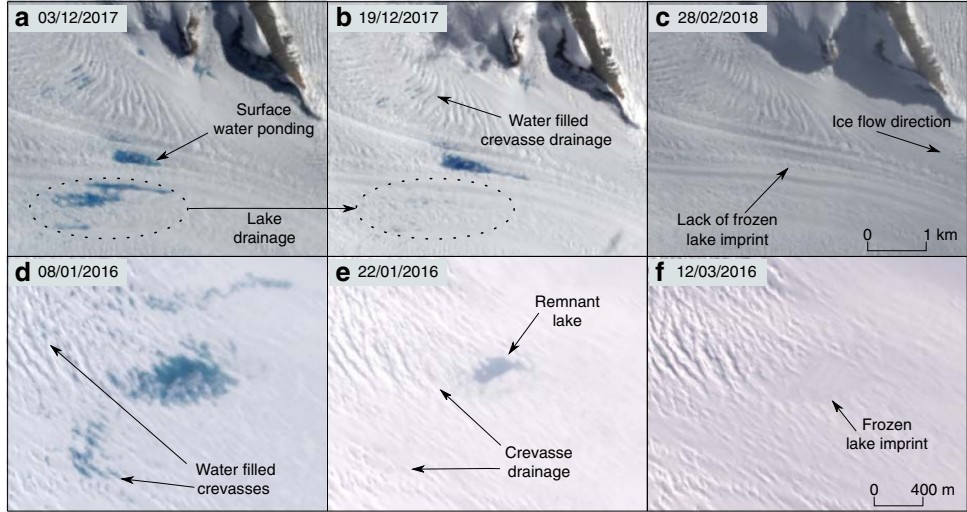

**Fig. 5** Satellite images of surface meltwater. **a–c** Landsat 8 images of ponds and water-filled crevasses draining on Hektoria Glacier, location is shown in Supplementary Fig. 9. **d**, **e** Sentinel 2 images of a region of potential lake and crevasse meltwater drainage, showing subsequent freezing on Drygalski Glacier. Location is shown in Supplementary Fig. 10

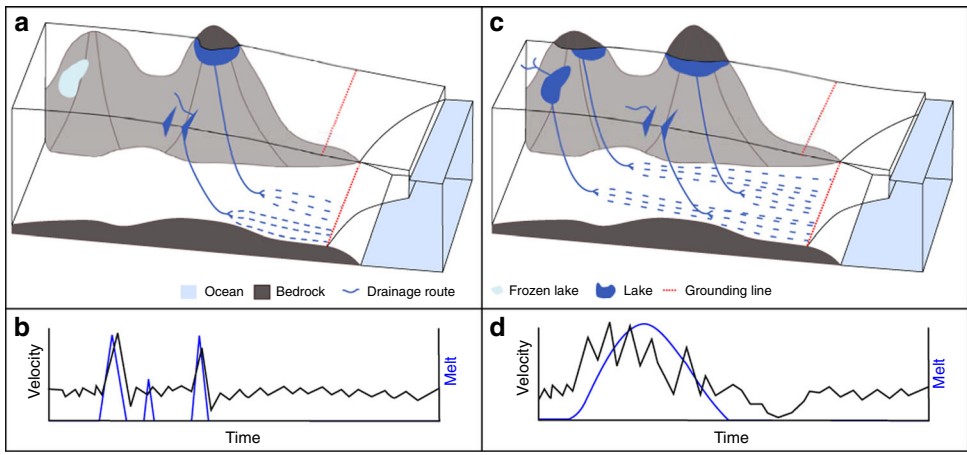

**Fig. 6** Schematic of potential shift in seasonal Antarctic glacier dynamics. **a** Current situation in Antarctic Peninsula. **b** Dynamics of Antarctic peninsula and High Arctic melt-induced speed-ups. **c** Greenlandic, and potential future Antarctic drainage systems. **d** Ice melt and speed response to scenario shown in **c**

meltwater drainage and the resulting dynamic response occurring within 6 days. We suggest that once at the ice base, this water causes a rapid acceleration of ice by promoting enhanced basal sliding through a spike in basal water pressure[4,9,22]. Efficient evacuation of both surface-derived meltwater and water stored at the ice base would then lead to a subsequent reduction in water pressure and ice motion[1,13,22]. This mechanism is known to occur in alpine glaciers[5], some polythermal glaciers[6,23], and across some marginal regions of the Greenland Ice Sheet[1,9], but this is the first time that surface meltwater-induced speed-up has been observed in Antarctica.

Antarctic Peninsula melt seasons are currently characterised by sporadic melt events interspersed with longer periods where surface melting does not occur (Fig. 2b). Spikes in modelled melt are concurrent with Föhn wind conditions[21], which are short-lived. As a result, the supraglacial hydrological network is not extensive, with few moulins and restricted growth of surface lakes; a configuration that favours refreezing rather than drainage of surface lakes (Fig. 6a). This low volume of surface-derived meltwater, and more restricted surface-to-bed hydraulic connections, along with the high dynamic sensitivity to surface meltwater input of Antarctic Peninsula glaciers demonstrated

here, suggests an inefficient subglacial hydrological system dominates for most of the year (Fig. 6b). The variable meltwater inputs to the bed may cause more efficient subglacial drainage to occur temporarily, but this is likely short-lived given that velocities rapidly return to pre-meltwater input magnitudes. Many High-Arctic glaciers are sensitive to surface meltwater inputs[25,26], which have the effect of inducing short-lived speed-ups superimposed on a seasonal melt season signal[7]. The subglacial hydraulic configuration, glacier dynamic response and lack of seasonal patterns in meltwater-induced velocity of Antarctic Peninsula glaciers creates a regime that is perhaps even more sensitive to meltwater inputs than that of High-Arctic glaciers.

Longer, more intense melt seasons are projected for the Antarctic Peninsula[15], with the potential to create a positive melt-elevation feedback as lakes grow and ice thins to expose bedrock[3] (Fig. 6b). Our observations indicate that ice-flow dynamics and atmospheric conditions are already coupled over short-timescales on the Antarctic Peninsula. We expect that this mechanism also operates in other parts of the Antarctic Ice Sheet that experience surface melting, and is likely to become more widespread, frequent and important for net annual ice motion if climate continues to warm. In the ablation area of the Greenland Ice Sheet,

prolonged surface melt sustains meltwater input into moulins[27] and leads to numerous surface lake drainages[28,29] each melt season. Under such conditions, melt-induced ice flow variations over sub-daily to seasonal timescales combine to create an annual velocity signal[1,30], reflecting the adjustment of the basal hydrological system to seasonal meltwater inputs[22,31]. Surface melt has also been shown to influence glacier velocity throughout the year in Alaska[32], with a strong positive correspondence between greater summer surface melting and winter velocity slow-down. This is a consequence of more efficient subglacial drainage during high summer melt seasons reducing the availability of subglacial water during the following winter. Increases in surface melting over the coming decades[15] could change Antarctic Peninsula glacier dynamics, with an increasing role for forcing by longer-term climatic change rather than individual weather events such as Föhn winds. We envisage that the current Antarctic Peninsula glacier regime will transition toward a High-Arctic regime (a seasonal signal with sensitive yet short-lived response to melt-water variations) and eventually toward a Greenlandic or Alaskan style meltwater-driven system of glacier flow[11,32] (Fig. 6c, d), whereby frequent and sustained speed-up events interact with the subglacial drainage system to regulate ice flow over seasonal timescales. The potential for such shifts caused by increasing melt is yet to be incorporated into numerical models used to predict the future mass balance of the Antarctic Ice Sheet and its contribution to sea level change[33] and future work should aim to determine the threshold volumes of melt required to generate these proposed shifts in glacier dynamics.

## Methods

**Ice velocity**. Ice velocity estimates were derived from feature and speckle tracking of Sentinel 1a and 1b Interferometric Wide Swath mode Single-Look Complex Synthetic Aperture Radar amplitude images. These data cover two swaths and include 184 6- or 12-day repeat image pairs between October 2016 and mid-April 2018.

Each image was split into many image patches, and cross-correlation of each image patch between repeat-pass image pairs was used to determine the offset of features (e.g., crevasses) over time[34]. Image pairs were initially co-located using precise satellite orbit ephemerides and converted to amplitude in GMTSAR[35,36]. A Butterworth high-pass spatial-frequency filter was then used to remove image brightness variations with a wavelength of greater than ~1 km[34], isolating movable surface features. Tracking of the co-located and filtered images was undertaken in MATLAB within PIVSuite (https://uk.mathworks.com/matlabcentral/fileexchange/45028-pivsuite) adapted for quantifying ice flow. Computationally efficient sub-pixel displacement estimates for each image patch were made by obtaining an initial estimate of the cross-correlation peak using a fast Fourier transform, and then up-sampling the discrete Fourier transform using matrix-multiplication of a small neighbourhood (1.5 × 1.5 pixels) around the original estimated location[37]. We oversampled the amplitude images in the azimuth direction by a factor of two[38] and used image patch sizes of 96 × 256 single-look oversampled azimuth and range pixels, with an overlap of 64 and 24 pixels respectively between patches. Correlation signal-to-noise ratios were used to filter the velocity results, with a threshold set at 5.8[35]. Spurious correlations which evaded this first sift were removed by a threshold strain filter[39], and a kernel density filter based on the paired displacements in the range and azimuth directions for each image patch[40]. A Visible Structured Noise Filter (VSNR)[41] was used to remove any anomalous stripes in the azimuth displacement data which sometimes result from fluctuating electron density along the sensor path through the ionosphere[38]. Noise in the azimuth displacement data was quantified using a Blind/Referenceless Image Spatial Quality Evaluator, and the VSNR filter was only employed if it helped to improve image quality (i.e., reduced striping). The filtered velocity fields were transformed from radar to map coordinates using the Advanced Spaceborne Thermal Emission and Reflection Radiometer Global Digital Elevation Model. The median velocity error was estimated to be 30.5 m a$^{-1}$ by measuring the mean velocity over bedrock areas.

Regions of interest (ROIs) 1 km by 1 km were drawn along the central flow lines of each glacier, allowing area-averaged velocity values to be extracted. The number and positioning of ROIs for each glacier was primarily dictated by velocity data coverage, with a minimum threshold of 80% required. ROIs were numbered sequentially inland from the grounding line, with each ROI indicating an approximate number of kilometres inland. We calculated velocities both for each ROI, and also a mean value for each glacier by averaging all the respective ROIs. A single chain of between six and ten ROIs were drawn for each glacier.

**Tidal modelling**. Tidal constituents for a location just off the east coast of the Antarctic Peninsula (−65.72 N, −61.12 W) were generated using the Oregon State University TOPEX/Poseidon Global Inverse Solution tide model TPXO[42]. This model is available at http://volkov.oce.orst.edu/tides/global.html. The Tide Model Driver package was run in MATLAB to produce modelled tidal amplitudes and frequencies for the study period from these constituents. These data were used as an approximation of the tidal signal for the whole study region. We calculated a continuous wavelet transform (using a standard Morse wavelet) of daily-averaged tidal amplitudes and the 6-day velocity data to investigate temporal correlations between the tides and the area-averaged ice velocity from each ROI for each glacier. The results of this analysis are plotted as a scalogram (Fig. 4). A scalogram is a visualisation of the absolute value of the continuous wavelet transform of a signal, plotted as a function of time (x-axis) and signal frequency (y-axis). The colours represent the strength of a signal of a particular frequency at a given point in time. The scalogram can be conceptualised as similar to a Fast Fourier Transform of a time-series, but one that retains information about the strength of signals of different frequencies at different times (or indeed distances) rather than just displaying a time-independent visualisation of the frequency of the strongest signals within a time-series.

**Satellite image analysis**. Ice front position and conditions were mapped using the tool GEEDiT[43] in Google Earth Engine ©. This analysis used cloud free Landsat 8 and Sentinel 2 optical imagery, as well as Sentinel 1 radar return images. In total 750 ice front margins were manually identified and mapped (Supplementary Fig. 11). Margin position change using was then defined using a extrapolated centre-line method[44] implemented in the MaQiT tool[43].

To look for possible routes of water getting to the ice-bed interface, visible waveband Landsat 4,5,7, and 8, and Sentinel 2 imagery was analysed to identify patterns of surface melt across the studied glaciers. GEEDiT was used to cycle through available imagery[43]. Images with a cloud cover of greater than 80% were excluded from the analysis. In many of the remaining images, cloud cover obscured the ice surface. We therefore extended our analysis beyond the period of time which velocity data were available, to that of the Landsat record (1984 onward). Where the ice surface was visible (in 600 instances), images were classified as to whether obvious signs of surface meltwater, in the form of surface lakes, meltwater-filled crevasses and streams, were present. Meltwater features were found to recur in the same positions on glacier surfaces, meaning the Supplementary Figs. 5–9 are indicative of typical inter-annual meltwater patterns.

**Regional climate modelling**. We used a high-resolution (5.5 km) version of the Regional Atmospheric Climate Model (RACMO) version 2.3p2 of the Antarctic Peninsula region[21]. The model run was initialised on 1st January 1979, with outputs from October 2016 to mid-April 2018 analysed here. The surface mass balance model has been validated against 132 observational records[21]. Mean values of snowmelt were extracted from the modelled snowmelt data in 1 km by 5 km areas along the centreline and extending inland from the grounding line of each glacier. These regions of analysis overlap with the ROIs used to generate velocity data. Data were extracted using MATLAB and ArcMap.

## Data availability

The data that support the findings of this study are available from https://figshare.shef.ac.uk/s/896c34d71a41caf5d03b

## Code availability

Image processing to derive velocity estimates was performed using GMSTAR (https://topex.ucsd.edu/gmtsar/)

and the following Matlab © functions: Particle Image Velocimetry framework: https://uk.mathworks.com/matlabcentral/fileexchange/45028-pivsuite

Normalised cross-correlation with same-sized images: https://uk.mathworks.com/matlabcentral/fileexchange/29005-generalized-normalized-cross-correlation

Sub-pixel cross-correlation peak determination: https://uk.mathworks.com/matlabcentral/fileexchange/18401-efficient-subpixel-image-registration-by-cross-correlation

Image segmentation filtering: https://uk.mathworks.com/matlabcentral/fileexchange/19084-region-growing

Visible Structured Noise Filter: https://www.math.univ-toulouse.fr/~weiss/Codes/VSNR/VNSR_VariationalStationaryNoiseRemover.html

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

## Acknowledgements

J.C.E. acknowledges a NERC independent research fellowship grant number NE/R014574/1. J.M.W. acknowledges financial contributions made by the Netherlands Organization for Scientific Research (grant 866.15.201) and the Netherlands Earth System Science Center (NESSC).

## Author contributions

A.J.S., P.A.T., J.C.E. and S.J.L. conceived the work. P.A.T. collected and analysed the velocity data with assistance from all authors. A.J.S. wrote the code to derive velocity estimates from Sentinel 1 data with assistance from B.J.D. J.C.E. analysed the Landsat and Sentinel 2 satellite imagery with assistance from J.H. J.M.W. provided the RACMO data. J.C.E. led the preparation of the manuscript with contribution from all authors.

## Competing interests

The authors declare no competing interests.
