## [Peer Review File · Nature Communications]

Reviewers' comments:

Reviewer #1 (Remarks to the Author):

Please see my attached PDF

Reviewer #2 (Remarks to the Author):

This paper considers velocity variations measured on five outlet glaciers in the Antarctic Peninsula. The conclusion is that these variations are driven by surface melt and not by processes acting at the glacier terminus or grounding line. This would be an important conclusion that goes against most of the views published these days.

According to the authors, "There is a high qualitative correspondence between periods of modelled surface melting and speed-up events" (l. 68-69). I do not find these correlations very convincing, especially when considering the Extended Data Figures 1-4.

ED Figure 1 shows a melt peak at (around?) April 17, which corresponds to a speed up at all four locations on Crane Glacier. The same is true for Jorum Glacier (ED Figure 2). However, on Jorum Glacier there is another peak in velocity at 4 and 5 km inland that precedes the April 17 melt peak. This preceding peak is not present in the velocity record for 1 km inland on Jorum Glacier. In late October or early November (add years to axes of ED Figure 1-4) the speed-up on Crane Glacier seems to precede the melt event. Further, on all glaciers considered, there are many speed-up events during the austral winter that do not appear to be correlated with melt events. Conversely, melt events present in especially ED Figure 4 do not appear to be linked with speed-up events. Perhaps a more quantitative correlation between speed up and melt events would be more convincing.

ED Figures 5-9 show in more detail some of the meltwater features on the different glaciers. If drainage of surface lakes results in more water at the bed, is there sufficient water input to the bed to cause significant increases in sliding speed? On a more elementary level, the authors argue that "currently, little water reaches the bed" (l. 19) which begs the question what causes the high speeds observed on these glaciers in the first place?

The variations in velocity are larger than the estimated uncertainty (the grey bands in ED Figures 1-4 could be made more prominent). Yet there seems to be some perturbing pattern in these velocities: every speed up is followed immediately by a slow down. There are no (or very few) periods of speed up or slow down occurring over more than one time interval over which velocities are determined. This may well be the case for the actual velocities, but requires some explanation – and also seems to argue against the meltwater hypothesis as driver of accelerations.

There are some interesting seasonal differences between the five glaciers studied. Crane Glacier shows a minimum speed around the end of the austral winter. Jorum Glacier shows a small peak in August followed by a slow down. Cayley Glacier shows an even greater velocity maximum somewhere in August. Finally, Hektor Glacier shows a seasonal velocity pattern similar to Crane Glacier. Why do these seasonal differences occur and what is/are the possible mechanism(s) for seasonal velocity variations?

I agree that the velocity events appear to be unrelated to what happens at the grounding line or glacier terminus. Nevertheless, the point could be strengthened by better explaining what is shown in Figure 3. What is a "scalogram"? This behavior is different from the Ross Ice Streams which show a tidal influence over considerable distances upglacier. Can the authors speculate what causes this different behavior?

The ROI for all glaciers stops at the grounding line (Figure 1). Why is this? Can the velocity measurements be extended to include the floating part to evaluate and perhaps investigate the cause of speed variations on the floating parts?

Finally, about the RACMO model: how well has that model been calibrated and how do we know that the melt rates (especially the timing of melt events) predicted by this model are actually correct?

This paper presents interesting and important data on seasonal variations in velocity and shorter-duration speed up and slow down events. The explanation for why these speed up events occur is appealing but at this stage not convincing to me.

Reviewer #3 (Remarks to the Author):

I think the results presented in the paper are interesting, and they are to my knowledge original. They do suggest that a new set of changes to mass balance and hydrological processes on Antarctic glaciers (and, in due course, potentially the whole ice sheet) may be imminent and that a transition to conditions more like those seen on glaciers in the High Arctic may be underway, at least in maritime Antarctica. What that will mean, in a global sense, requires a lot more thought, however, than is apparent here. This manuscript takes the first step in exploring this issue, but doesn't really follow through in the logical, coherent, and compelling way that would be needed to convince a broad readership that something very fundamental is about to change in terms of Antarctic ice sheet dynamics.

Detailed Comments for Authors (keyed by line number as in the Authors' manuscript):

16: respond rapidly to...

18: but that rapid and efficient...

21-23: Do climate models actually project "longer and more intense melt seasons" for this region - if they do it would be helpful to say something about the magnitude of changes that have been projected and the timescales on which they might be expected to occur

28: on sub-daily to decadal timescales

28-29: I think you mean "over a range of timescales down to the sub-diurnal level"

30: How relevant do you think climate change forcing might be at sub-diurnal timescales?

32: On the Antarctic peninsula

33-34: How and why does surface meltwater "enhance" rapid ice shelf disintegration?

38: on the Antarctic Peninsula

39: during the period October 2016 to April 2018

40: while Cayley Glacier is on the west side

42: extending inland from the glacier's grounding line

43: accelerations in ice velocity to values more than 20% greater than....

46: the largest event occurred at Drygalski Glacier in November 2017 (Figure 2a), when velocities...

48: recorded in individual ROIs

51: Most of the events were....slow-down to velocities below pre-event values..

52: not obvious to me what "of a similar duration but smaller magnitude" means in this context

53: have not previously been reported from Antarctica

54: in ROIs further...

55: opposite of what would be expected if...

56: or iceberg calving events

58: of the velocity data

59: but that they diminish ...

61-62: no role for marine processes - given the limited data (tide data) presented this seems like a bit of a stretch

63: whether there is a temporal correspondence between

65-66: melt season lasts from October to April, and a small....event is modelled in July 2017.

68: modelled, separated by periods of little...

72-74: also consistent with much of what we know about the dynamics of valley glaciers in alpine regions.

82: not obvious to me how you can make deductions about the magnitude of horizontal stress gradients from crevasse patterns. In any case, isn't the important factor the presence of the crevasses that the water can flow into? So, it's not clear to me why you need to speculate about stress gradients anyway.

87-89: Would be useful to say something about what is generating these melt events - is it just the crossing of some temperature threshold during the seasonal cycle or are they connected to specific weather events - and if so, what is the nature of these events? Are the same events affecting glaciers on either side of the Peninsula - and, if they are, are the responses simultaneous or lagged?

95: and ice motion

96: occur in alpine glaciers...

98: I would say "observed" rather than inferred - otherwise you are signalling a certain lack of confidence in what you are saying

102-103: It seems to me there is a significant logical leap embedded in this statement. The meltwater drainage systems that are inferred to exist in this study are very similar to those that occur on many polythermal glaciers across the Arctic and yet those glaciers are not, in general, making a headlong rush for the ocean as a result of water reaching their beds every summer. For sure their flow is influenced by temporal variability in water inputs and subglacial water pressure/storage, and we might expect to see this become more obvious in Antarctica as a result of the sorts of changes that are described in this manuscript. I agree that this may require some shift in how we model Antarctic outlet glacier dynamics going forward and that there is some potential for dynamic changes and morphological adjustments to such changes. So, it seems fine to point that out, but I'm not sure that it's very helpful to speculate on what these adjustments will be. In general, it seems to me that glacier response to warm weather events (which is really what is described in this paper) will not necessarily be identical to glacier response to secular climate forcing - simply because a whole range of additional processes (like changes in firn stratigraphy, facies zone distribution etc.) start to matter on longer (climatological) time scales.

Incidentally, I have a feeling that the glaciers of the Canadian High Arctic may be a better analogue for the end-point of the transition that the authors imagine is now underway in Antarctica than the glaciers of maritime west Greenland with which they are familiar. However, I recognise that west Greenland may become a better analogue further down the road if atmospheric warming continues. The authors should perhaps consider that possibility when building a scenario for how conditions in Antarctica may evolve in the medium to long term.

Martin Sharp

All line numbers in our response refer to the revised manuscript unless otherwise stated.

Reviewer #1 (Remarks to the Author):

Summary

Main point 1. This paper attempts to show meltwater induced speed-up of Antarctic grounded glaciers, which is a well-documented (and very important) process in Greenland, but has yet to be documented in Antarctica. However, unfortunately I do not think there is enough evidence to support this claim. Although the authors do show a qualitative correlation between ice velocities and modeled surface melting, this correlation could simply be an indirect result of warmer air temperatures, which may have caused the ice shelf or land-fast sea ice to break up, thereby resulting in short-term increases in upstream ice velocities.

We thank reviewer 1 for their comments on the manuscript. Reviewer 1's main concern is that ice-shelf or sea-ice processes are the cause of the speed-up events we observe. To investigate this, we analysed the satellite record to quantify sea ice and ice shelf front position changes and conditions during our study period (Lines 78-89; Extended Data Figures 5,6 and 7). This led us to conclude that neither shelf nor sea ice processes are the likely cause of speed-ups for the following reasons:

1) Frontal position remained remarkably stable during the study period (Line 80-82; Extended Data Figures 5 and 6), with the only clear pattern being the minor (<500 m) advance of two glaciers (Line 83). These advances are the opposite of what we would expect if marine processes controlled the speed-up events.

2) Large iceberg calving events did not occur during the speed-up events. All glaciers were dominated by small-calving events.

3) The characteristics of speed-up events, with increased relative magnitude upglacier and short duration, are inconsistent with either sea-ice or shelf processes (Lines 65-73; Figure 3). Both sea-ice and shelf processes remove back-stress on the glacier, which would create a prolonged (seasonal in the case of sea-ice break up and reformation) glacier acceleration, or even a velocity step change, initiated at the margin and propagating upstream (e.g. Moon et al. 2015).

We therefore retain our original conclusion, based on the evidence that we present, that the short-lived but high magnitude speed-up events (the definition of which we have clarified in the text) are caused by surface meltwater reaching the base of the studied glaciers. This is evidenced by the characteristics of the speed-ups (increasing in magnitude upstream), co-occurrence with periods of modelled melting and observations of surface drainage occurring on the glaciers.

Main point 2. Although the authors do discuss the potential for the observed speedups being caused by marine processes such as tidal variations, calving, and ice break up, they rule those processes out on the basis that the "speed-up events become relatively larger at ROIs further from the marine margin" (line 54/55). But in fact, according to Fig 2b, these speed-up events actually decrease upglacier. So all of the explanation associated with this seems to be flawed.

This point seems to be rooted in a miscommunication as to our meaning of "relative magnitude" of speed-up events. While it is true that the *absolute* magnitude of the speed-up events decreases upglacier (as does the background ice velocity), the *relative* magnitude of speed-up events (that is the velocity variation during a speed-up event in comparison to background mean) tends to increase upglacier for those events associated with meltwater spikes. This was plotted on Figure 4 (Figure 3 in original submission), but was insufficiently signposted. We now outline this in the caption of Figure 4 and include a further figure, which shows the relative magnitude speed-up events increasing upglacier for the most prominent speed-up events (Figure 3).

Main point 3. To improve, I suggest that the study is expanded to include at least one land terminating glacier (if possible) where marine processes that could influence the ice velocities would not be present. Additionally, a detailed analysis of satellite imagery showing the glacier fronts should be undertaken to try to establish the dynamics of the ice shelves and sea ice, and therefore their possible influences on upstream glacier velocities. If such analysis were to be done, I would then look forward to reading a revised manuscript.

Unfortunately, land-terminating glaciers do not exist in our study area (a consequence of the low equilibrium line altitude on the Antarctic Peninsula). Instead, we have undertaken extensive analysis of satellite imagery to analyse front position and conditions of the marine-terminating glaciers included in our study. The results are summarised in our first response to this reviewer and demonstrate that shelf and sea-ice processes are not the cause of the observed speed-up events. In addition, we assessed the possibility that tides drive the variations in velocity (e.g. see Figure 4). The results of the scalogram for Hektor Glacier (see reply below regarding making this clearer to follow) indicate that smaller scale background fluctuations in ice velocity close to the grounding line could be controlled by tides, but that there is also a velocity signal that becomes relatively larger (compared to the background ice motion) away from the sea up to 8 km inland and this corresponds with a meltwater spike.

In general, given the long list of authors on this paper, many of who are relatively senior, I was surprised to have to make some of the comments above (and below), especially the one I make two paragraphs above. Some parts of the paper also do not seem to have been proof read adequately (e.g. I cannot make sense of Figure 2's caption).

We apologise for mis-labelling b and c on Figure 2, making the caption difficult to comprehend.

Please see below for additional line by line comments.

Line by line comments

11: 'ice dynamics' is vague, especially as we already know that warming = increased melting = potential ice shelf break up and therefore speedup of upstream glaciers. Clarify that you're referring to dynamics at the ice bed interface.

This is now clarified in the text (Line 11).

11 -15: Having read the rest of the paper, I don't think you have enough evidence to say this (for the reasons stated in my summary above).

Our data show large and rapid speed-ups occur. Given that we show that tidal, sea-ice and ice-shelf processes do not cause these, and that there are multiple lines of evidence that support meltwater being the trigger for speed-ups (Lines 132-135), we retain the original wording. See our response to main point 1 above for further evidence.

16 – 23: Far too much speculation for an abstract.

We are unsure which part of the abstract is deemed too speculative. We believe this abstract summarises our paper succinctly and accurately.

25/26: As this is a key statement, on which the premise of the rest of the study is based, I think some more Greenland-based observational as well as modelling studies should be added, e.g. Das et al (2008), Tedesco et al (2012) (both observational), and Bougamont et al (2014) and Banwell et al (2016) (both modelling).

Wider reference to the literature has been added as suggested by the reviewer (Line 29).

27: 'ice base' is vague, as it could mean the underside of an ice shelf...

Rephrased (Line 28).

38: It would have been very useful to also analyse at least one land terminating glacier, i.e. which cannot be influenced by the break-up of an ice shelf or land-fast sea ice.

Unfortunately, this is not possible at this study location due to the low regional equilibrium line altitude (response to main point 3). However, we have now conducted extensive analysis of glacier front conditions (Extended Data Figures 5 and 6).

43: "rapid accelerations in ice velocity" over what timescales.

The timescales were noted a few lines later (lines 58-59 in adapted manuscript, 50-51 in the original submission).

54/55: Fig 2c actually shows that the speed decreases at ROIs further from the marine margin... So the rest of this discussion in this paragraph seems to be flawed.

We have clarified our definition of *relative* magnitude of speed-up events in our response to main point 2 and added an additional figure plotting the relative magnitude (Figure 3).

68 – 70: An alternative possibility for this correlation between glacier speed up and surface melting is that the higher air temperatures (i.e. that contributed to the increased melting) also acted to break-up parts of the ice shelves and/or sea ice, which would also cause upstream glacier acceleration.

See our response to main points 1 and 3 above

75: Add 'help' (or similar) in between 'To' and 'identify'. As optical imagery analysis is not going to enable you to know with 100% confidence whether melt is getting to the bed or not.

Added (Line 117).

78: Describe how the lakes drained – did they overflow, or do you think there is evidence that they drained through their bottoms? (i.e. possibly via hydrofracture). Another alternative that should be mentioned is whether they may have simply drained into firn (either below or laterally, perhaps firn via a surface stream).

We now discuss the observations of surface lake drainage in greater detail (Lines 118-130). The possibility that some water gets into the firn but does not reach the bed is now also mentioned (Line 129).

80: yes, stream terminations may indicate moulins (or simply crevasses? Or even low density firn?), but it seems there is no evidence that those moulins go to the bed?

The possibility of surface lake drainage into the firn (and no further) is now mentioned (Line 129). We now also label all stream terminations as "potential" moulins. We have changed the word "likely" to "potential" routes to the bed (Line 129).

87-92: I don't think this statement can be made without having made observations at the front of each glacier to see what the ice shelves and sea ice were doing during the study period. Including a land terminating glacier in the analysis would also be very useful.

Observations of ice front of each glacier now included (Extended Data Figures 5 and 6). These do not show a correspondence with the timing of speed-up events.

93 – 95: You have no evidence for this.

This interpretation is based on our current understanding of glacial hydrology, including multiple studies of similar systems in Greenland (e.g. Sole et al., 2013; Zwally et al., 2002; Banwell et al., 2016; Das et al., 2008; Tedstone et al., 2015; Schoof, 2010), the Arctic (Wyatt and Sharp, 2015; Pimentel et al., 2017) and alpine settings (Iken and Bindschadler, 1986).

Figures

Fig 1: What are the black boxes in b – f?

The black boxes now explained.

Fig 2: I have read the caption many times and I can't work out what is in b) or c). The caption is extremely unclear. E.g. the caption suggests that b) is modeled melt, but then the y-axis title is 'speed'? And the caption suggests that c) is for separate ROIs, but which is which glacier in the plot? Also, in b) the speed of glacier decreases away from the ice front, NOT increases, as the text says. I also think that c) and d) could be combined, as the same data (the black line) is shown in both.

We apologise for the mislabelling of b and c in Figure 2. This explains the reviewer's difficulty in interpreting the caption. Subplots b and d (previously c and d) do not show the same data. B is for the entire region, d is for a single glacier, hence the difference in y-axis values.

References

Banwell, A., I. Hewitt, I. Willis, and N. Arnold (2016), Moulin density controls drainage development beneath the Greenland ice sheet, *J. Geophys. Res. Earth Surf.*, 121, 2248–2269, doi:10.1002/2015JF003801.

Bougamont, M., P. Christoffersen, A. L. Hubbard, A. A. Fitzpatrick, S. H. Doyle, and S. P. Carter (2014), Sensitive response of the Greenland ice sheet to surface melt drainage over a soft bed, *Nat. Commun.*, 5, 5052–5052, doi:10.1038/ncomms6052.

Das, S. B., I. Joughin, M. D. Behn, I. M. Howat, M. A. King, and D. Lizarralde (2008), Fracture propagation to the base of the Greenland ice sheet during supraglacial lake drainage, *Science*, 320, 778–781, doi:10.1126/science.1153360.

Tedesco, M., I. C. Willis, M. J. Hoffman, A. F. Banwell, P. Alexander, and N. S. Arnold (2013), Ice dynamic response to two modes of surface lake drainage on the Greenland ice sheet, *Environ. Res. Lett.*, 8(3), 34,007, doi:10.1088/1748-9326/8/3/034007

Reviewer #2 (Remarks to the Author):

This paper considers velocity variations measured on five outlet glaciers in the Antarctic Peninsula. The conclusion is that these variations are driven by surface melt and not by processes acting at the glacier terminus or grounding line. This would be an important conclusion that goes against most of the views published these days.

According to the authors, “There is a high qualitative correspondence between periods of modelled surface melting and speed-up events” (l. 68-69). I do not find these correlations very convincing, especially when considering the Extended Data Figures 1-4.

We thank reviewer 2 for their comments on the manuscript, and we are glad that they find that we have drawn an “important conclusion”. We hope that our revisions are more convincing as to the cause of the speed-up events. Importantly, we have clarified which velocity fluctuations we refer to as

surface meltwater drainage-induced “speed-up events” (Lines 47-50) and described why we would not expect a simple linear pattern between surface melt magnitude and speed-up (Lines 105-113).

ED Figure 1 shows a melt peak at (around?) April 17, which corresponds to a speed up at all four locations on Crane Glacier. The same is true for Jorum Glacier (ED Figure 2). However, on Jorum Glacier there is another peak in velocity at 4 and 5 km inland that precedes the April 17 melt peak. This preceding peak is not present in the velocity record for 1 km inland on Jorum Glacier.

The preceding peak on April 17 at Jorum is not related to a spike in meltwater and is therefore unlikely to be triggered by the drainage of surface meltwater to the bed. We certainly do not claim in this paper to have an explanation for each individual velocity fluctuation; our focus is on the association between the largest short-lived velocity variations that tend to increase in relative magnitude (i.e. compared to background ice motion) inland and their association with periods of meltwater formation on the surface of the glaciers. The event the reviewer refers to is an interesting spike as it is larger than the background (probably tidal) variations and so may have been caused by a different mechanism, e.g. a subglacial lake drainage which we cannot detect using our analysis of optical satellite imagery. However, we consider further investigation of this to be outside the scope of our study.

In late October or early November (add years to axes of ED Figure 1-4) the speed-up on Crane Glacier seems to precede the melt event.

Our data have a temporal resolution of 6 days due to the combined repeat period of Sentinel 1 a and b (12 days each offset by 6 days). This apparent offset is within this 6 day window either side. Years are already included on the axes of ED figures.

Further, on all glaciers considered, there are many speed-up events during the austral winter that do not appear to be correlated with melt events.

We have labelled the events we are referring to as speed-up events with dotted lines on the figures and defined them in the text (Lines 50-51; Extended Data Figures 1-4). Velocity fluctuations do occur outside of these, and we are not disputing this, but many are small compared to the speed-up events we are interested in and/or likely to be related to the influence of tides (e.g. see Fig. 3 and corresponding discussion in the text) or other processes that we cannot detect using the analysis presented (such as the drainage of a subglacial lake). These fluctuations are smaller than the speed-up events with which this paper is interested and also do not show a characteristic increase in relative velocity away from the grounding line (Figure 3; Extended Data Figures 1-4).

Conversely, melt events present in especially ED Figure 4 do not appear to be linked with speed-up events. Perhaps a more quantitative correlation between speed up and melt events would be more convincing.

It would not be expected that every spike in melt rates would cause a speed-up event. Indeed, theoretical and observational studies have shown that it is high melt variability (rather than total melt amount) that limits the ability of the subglacial hydrological system to adjust to water inputs, creating a more sensitive ice velocity response (e.g. Schoof et al., 2010). Therefore, we would only expect speed-ups during large, rapid melt events or melt events preceded by little melt (where subglacial drainage is inefficient). This is indeed the case, and we have now expanded upon this in the text (Lines 104-110). For example, at the beginning of the melt season (November 2016) Hektorica responds very sensitively to a relatively small meltwater spike (and again possibly in July 17). Further meltwater spikes of similar magnitude after this initial melt event do not elicit a similar velocity response (e.g. the period between January 2017 and March 2017), presumably as the system has evolved to cope with the meltwater input (Extended Data Figure 4). Given this highly non-linear

response of glaciers to supraglacial melt drainage to the bed (modulated by subglacial hydrology), a simple quantitative correlation between surface melt and ice velocity is not expected (Schoof et al., 2010), nor has it been identified in studies conducted in e.g. Greenland (e.g. Sole et al., 2011; Bougamont et al., 2014).

ED Figures 5-9 show in more detail some of the meltwater features on the different glaciers. If drainage of surface lakes results in more water at the bed, is there sufficient water input to the bed to cause significant increases in sliding speed? On a more elementary level, the authors argue that “currently, little water reaches the bed” (l. 19) which begs the question what causes the high speeds observed on these glaciers in the first place?

Subglacial water systems can be highly sensitive to extra surface-derived water input. The volume of a small surface lake (1 m deep, 1000 m long and wide = $1 \times 10^6 \text{ m}^3$) is likely to be an order of magnitude greater than the sum of basal melting assuming that it occurs across the entire glacier base (basal melting typically on the order of 0.001 m/a (Fowler, 2010) over a glacier 20,000 m long and 5000 m wide = $1 \times 10^5 \text{ m}^3$). Furthermore, water from a surface lake drainage has the potential to reach the bed over short timescales, whereas basal melt values are based on an annual mean water flux. By “little water reaches the bed” we were speaking relative to the High-Arctic and Greenland comparison, where most research on this topic has been done. This is now rephrased in the manuscript (Line 20).

The variations in velocity are larger than the estimated uncertainty (the grey bands in ED Figures 1-4 could be made more prominent). Yet there seems to be some perturbing pattern in these velocities: every speed up is followed immediately by a slow down. There are no (or very few) periods of speed up or slow down occurring over more than one time interval over which velocities are determined. This may well be the case for the actual velocities, but requires some explanation – and also seems to argue against the meltwater hypothesis as driver of accelerations.

Slow-downs below the pre-event velocity are characteristic of meltwater driven speed-ups (e.g. Sole et al., 2011). These slow-downs are thought to be related to the drop in basal water pressure, and thus reduced basal sliding, caused by the transient increase in basal hydraulic efficiency, and evacuation of water stored subglacially, following the sudden influx of surface-derived meltwater. Our velocities are determined over 6-day periods due to the repeat time of Sentinel 1a and 1b. Melt events are typically shorter than 6 days, so we would not expect the resultant speed-up and slow-down to last for longer intervals.

There are some interesting seasonal differences between the five glaciers studied. Crane Glacier shows a minimum speed around the end of the austral winter. Jorum Glacier shows a small peak in August followed by a slow down. Cayley Glacier shows an even greater velocity maximum somewhere in August. Finally, Hektor Glacier shows a seasonal velocity pattern similar to Crane Glacier. Why do these seasonal differences occur and what is/are the possible mechanism(s) for seasonal velocity variations?

Whilst interesting, these seasonal variations are not the focus of this paper. We have clarified in the revised manuscript what we term as speed-up events, and isolated them from the speed-up events we are interested in (lines 47-50). It is likely that they occur due to seasonal-scale processes, perhaps marine based as they diminish further from the grounding line.

I agree that the velocity events appear to be unrelated to what happens at the grounding line or glacier terminus. Nevertheless, the point could be strengthened by better explaining what is shown in Figure 3. What is a “scalogram”? This behavior is different from the Ross Ice Streams which show a tidal

influence over considerable distances upglacier. Can the authors speculate what causes this different behavior?

A scalogram is a visualisation of the absolute value of the continuous wavelet transform of a signal, plotted as a function of time (x-axis) and signal frequency (y-axis). The colours represent the strength of a signal of a particular frequency at a given point in time. The scalogram can be conceptualised as similar to a Fast Fourier Transform of a time-series, but one that retains information about the strength of signals of different frequencies at different times (or indeed distances) rather than just displaying a time-independent visualisation of the frequency of the strongest signals within a time-series. Scalograms are used extensively in signal processing and are especially useful when analysing signals characterised by quasi-regular variability punctuated by abrupt changes (such as those studied here), which are localised in both frequency and time. The scalogram is now also explained in the caption for Figure 3 and in the Methods (lines 365-372). It is possible that the difference in response to tidal variations between our glaciers and the Ross ice streams is caused by the Ross streams having lower surface slopes (and thus being close to the floatation thickness for a greater distance upglacier) and larger ice shelves. The glaciers included in our study are also mostly bounded at their lateral margins by bedrock walls. Thus, it seems likely that more of the resistance to ice flow is provided by lateral drag that would not be affected by tidal height changes of the ice shelf.

The ROI for all glaciers stops at the grounding line (Figure 1). Why is this? Can the velocity measurements be extended to include the floating part to evaluate and perhaps investigate the cause of speed variations on the floating parts?

The ice shelves are quite short (a few 100s of m) so there is limited opportunity to study their behaviour. The melange in front of the shelves usually deforms considerably between radar images, which leads to loss of coherence between images and therefore false correlations (and anomalous velocity data) that are removed by our post-processing routine.

Finally, about the RACMO model: how well has that model been calibrated and how do we know that the melt rates (especially the timing of melt events) predicted by this model are actually correct?

The mass-balance component of RACMO has been compared to all available observations in the region, against which it performs well. This is now mentioned in the text (Line 394).

This paper presents interesting and important data on seasonal variations in velocity and shorter-duration speed up and slow down events. The explanation for why these speed up events occur is appealing but at this stage not convincing to me.

Reviewer #3 (Remarks to the Author):

I think the results presented in the paper are interesting, and they are to my knowledge original. They do suggest that a new set of changes to mass balance and hydrological processes on Antarctic glaciers (and, in due course, potentially the whole ice sheet) may be imminent and that a transition to conditions more like those seen on glaciers in the High Arctic may be underway, at least in maritime Antarctica. What that will mean, in a global sense, requires a lot more thought, however, than is apparent here. This manuscript takes the first step in exploring this issue, but doesn't really follow through in the logical, coherent, and compelling way that would be needed to convince a broad readership that something very fundamental is about to change in terms of Antarctic ice sheet dynamics.

We would like to thank reviewer 3 for a very useful and constructive review. We are glad that the reviewer acknowledges our interesting and original observations. The suggestion of comparing our

results to those in the high Arctic was particularly useful - our results from Antarctica are very similar to this Arctic situation. To increase the impact for a broad readership, we have expanded the section where we discuss possible future changes in Antarctic ice sheet dynamics (Final paragraph of the main text).

We have made all the minor changes to the text suggested by the reviewer below and provide a response to the more substantive changes.

Detailed Comments for Authors (keyed by line number as in the Authors' manuscript):

16: respond rapidly to...

18: but that rapid and efficient...

21-23: Do climate models actually project "longer and more intense melt seasons" for this region - if they do it would be helpful to say something about the magnitude of changes that have been projected and the timescales on which they might be expected to occur

We have added more details in the main text (Line 34-36) and altered this statement in the abstract.

28: on sub-daily to decadal timescales

28-29: I think you mean "over a range of timescales down to the sub-diurnal level"

30: How relevant do you think climate change forcing might be at sub-diurnal timescales?

We have rephrased this sentence.

32: On the Antarctic peninsula

33-34: How and why does surface meltwater "enhance" rapid ice shelf disintegration?

We have added an explanatory clause to this sentence.

38: on the Antarctic Peninsula

39: during the period October 2016 to April 2018

40: while Cayley Glacier is on the west side

42: extending inland from the glacier's grounding line

43: accelerations in ice velocity to values more than 20% greater than...

46: the largest event occurred at Drygalski Glacier in November 2017 (Figure 2a), when velocities...

48: recorded in individual ROIs

51: Most of the events were....slow-down to velocities below pre-event values..

52: not obvious to me what "of a similar duration but smaller magnitude" means in this context

53: have not previously been reported from Antarctica

54: in ROIs further...

55: opposite of what would be expected if...

56: or iceberg calving events

58: of the velocity data

59: but that they diminish ...

61-62: no role for marine processes - given the limited data (tide data) presented this seems like a bit of a stretch

We have extended this analysis, looking at glacier frontal position, due to the comments of reviewer 1. Although we observe some changes in front position, sea ice conditions and shelf extent (Lines 88-90; Extended Data Figures 5 and 6), these do not co-occur with the speed-up events discussed in the manuscript. In addition, the patterns of change observed (e.g. numerous small calving events, frontal

retreat) are not consistent with the short-lived, large speed-up events. We therefore maintain that marine processes are not the driver of the speed-up events discussed in the manuscript.

63: whether there is a temporal correspondence between

65-66: melt season lasts from October to April, and a small....event is modelled in July 2017.

68: modelled, separated by periods of little...

72-74: also consistent with much of what we know about the dynamics of valley glaciers in alpine regions.

Reference to alpine glaciers added (Line 142).

82: not obvious to me how you can make deductions about the magnitude of horizontal stress gradients from crevasse patterns. In any case, isn't the important factor the presence of the crevasses that the water can flow into? So, it's not clear to me why you need to speculate about stress gradients anyway.

Agreed. Sentence removed.

87-89: Would be useful to say something about what is generating these melt events - is it just the crossing of some temperature threshold during the seasonal cycle or are they connected to specific weather events - and if so, what is the nature of these events? Are the same events affecting glaciers on either side of the Peninsula - and, if they are, are the responses simultaneous or lagged?

We do not see any time lag between melt events and velocity response in our data. However, our velocity data have a temporal resolution of 6 days, so it is possible that a lag of less than 6-days could be present, but we cannot discern it. If there is a lag, it is likely to be <6 days. This comment promoted evaluation of the model data to find the cause of melt periods. We now highlight the meteorological conditions (Fohn winds) that lead to melting and the potential consequences of this in the manuscript (Lines 147,175-177).

95: and ice motion

96: occur in alpine glaciers...

98: I would say "observed" rather than inferred - otherwise you are signalling a certain lack of confidence in what you are saying

102-103: It seems to me there is a significant logical leap embedded in this statement. The meltwater drainage systems that are inferred to exist in this study are very similar to those that occur on many polythermal glaciers across the Arctic and yet those glaciers are not, in general, making a headlong rush for the ocean as a result of water reaching their beds every summer. For sure their flow is influenced by temporal variability in water inputs and subglacial water pressure/storage, and we might expect to see this become more obvious in Antarctica as a result of the sorts of changes that are described in this manuscript. I agree that this may require some shift in how we model Antarctic outlet glacier dynamics going forward and that there is some potential for dynamic changes and morphological adjustments to such changes. So, it seems fine to point that out, but I'm not sure that it's very helpful to speculate on what these adjustments will be. In general, it seems to me that glacier response to warm weather events (which is really what is described in this paper) will not necessarily be identical to glacier response to secular climate forcing - simply because a whole range of additional processes (like changes in firn stratigraphy, facies zone distribution etc.) start to matter on longer (climatological) time scales.

Incidentally, I have a feeling that the glaciers of the Canadian High Arctic may be a better analogue

for the end-point of the transition that the authors imagine is now underway in Antarctica than the glaciers of maritime west Greenland with which they are familiar. However, I recognise that west Greenland may become a better analogue further down the road if atmospheric warming continues. The authors should perhaps consider that possibility when building a scenario for how conditions in Antarctica may evolve in the medium to long term.

We added the situation currently observed in Antarctica could change to one that is similar to that observed in the Canadian Arctic (thank you to the reviewer for pointing us to the literature) (Lines 156-158). We have rephrased the sentence on response to “climatic warming” to note that shorter atmospheric events are important currently (Lines 167-168; Lines 175-177), but maintain that melting may become more important with climate change. As the reviewer suggests, the Arctic situation is a key intermediate stage between current conditions in the Antarctic Peninsula and those of the Greenland Ice Sheet. We have now incorporated this into the text (Lines 177-181). However, we maintain that a Greenland-style system, where melting effects flow over the year, may eventually occur if surface melting continues to increase.

Martin Sharp

Additional references

Fowler, A.C., 2010. The formation of subglacial streams and mega-scale glacial lineations. *Proceedings of the Royal Society A: Mathematical, Physical and Engineering Sciences*, 466(2123), pp.3181-3201.

REVIEWERS' COMMENTS:

Reviewer #1 (Remarks to the Author):

I congratulate the authors for doing an excellent job in dealing with, in particular, my previous concerns, but also the concerns of the other two reviewers. I think that this now a very high quality paper and should be published without delay.

Reviewer #3 (Remarks to the Author):

The authors have responded positively and constructively to previous comments, improving the readability of the manuscript as a result. I think it is now more or less in a publishable form, but provide a few comments below that would improve clarity of intent and readability still further. They should be easy to implement.

Comments keyed by line number in authors' revised m/s.

49: m a-1

54: you use both yr-1 and a-1 in this m/s. Choose one form and stick with it.

57: just refer to them as events you have already been clear that you are talking about events in which the surface velocity increases

60: Do slow-down events typically follow speed-up events in the same area ? Or do you get slow-down events that are not obviously compensating for prior speed-up events? My sense is that you don't but it would be useful to be clear about this

61: This seems to suggest that the speed up events are progressively increasing the surface velocity in the areas where they occur. is that correct?

65: I think you mean surface meltwater drainage into the body of the glacier. make this clear.

73: this is the opposite of what would be expected if...

74: periodicity of the velocity variations

77: large glacier speed-up events, which occur beyond..

84: but there was no clear relationship between the timing of speed-up events and changes in glacier terminus position

89: "marine processes" is pretty vague. Can you be more explicit about what you are referring to here?

117: help to determine whether

119: Do any of the drainage events occur when the lakes are still frozen? We do see this in Arctic Canada where such events leave behind bergs made of fractured lake ice on the dry lake floor

166: in other parts

174-75: I don't understand this part of the sentence. Do you mean that winter velocities tend to

be low after high melt summers. presumably this is because it takes longer to shut down the subglacial drainage system after a high melt summer?

176: role for forcing by.....

183: predict the future mass balance of the Antarctic Ice Sheet and its contribution to sea level change.

318: swaths

Martin Sharp, University of Alberta

Response to reviewers – Tuckett et al., NCOMMS-19-02418A

Reviewer #1 (Remarks to the Author):

I congratulate the authors for doing an excellent job in dealing with, in particular, my previous concerns, but also the concerns of the other two reviewers. I think that this now a very high quality paper and should be published without delay.

Thank you to reviewer #1 for their comments which greatly improved the manuscript.

Reviewer #3 (Remarks to the Author):

The authors have responded positively and constructively to previous comments, improving the readability of the manuscript as a result. I think it is now more or less in a publishable form, but provide a few comments below that would improve clarity of intent and readability still further. They should be easy to implement.

Thank you to reviewer #3. The comments have greatly improved the manuscript, as has the generously given ideas.

Comments keyed by line number in authors' revised m/s.

49: m a-1

54: you use both yr-1 and a-1 in this m/s. Choose one form and stick with it.

Corrected to a^{-1} throughout.

57: just refer to them as events you have already been clear that you are talking about events in which the surface velocity increases

Corrected.

60: Do slow-down events typically follow speed-up events in the same area ? Or do you get slow-down events that are not obviously compensating for prior speed-up events? My sense is that you don't but it would be useful to be clear about this

We have added a sentence on this to clarify (lines 74 to 75 in corrected manuscript).

61: This seems to suggest that the speed up events are progressively increasing the surface velocity in the areas where they occur. is that correct?

We have clarified that ice velocities return to pre-event values after a cycle of speed-up slow-down events (line 74).

65: I think you mean surface meltwater drainage into the body of the glacier. make

this clear.

Clarified (Line 81)

73: this is the opposite of what would be expected if...

74: periodicity of the velocity variations

77: large glacier speed-up events, which occur beyond..

84: but there was no clear relationship between the timing of speed-up events and changes in glacier terminus position

The above changes have been made.

89: "marine processes" is pretty vague. Can you be more explicit about what you are referring to here?

We have rephrased this to highlight the marine processes we investigated (lines 105-106).

117: help to determine whether

Corrected.

119: Do any of the drainage events occur when the lakes are still frozen? We do see this in Arctic Canada where such events leave behind bergs made of fractured lake ice on the dry lake floor

We have no conclusive evidence of this currently. Our hunch is that some subsurface lakes have the potential to drain to the bed. We hope to gather more definitive evidence of whether this happens in the future.

166: in other parts

Corrected.

174-75: I don't understand this part of the sentence. Do you mean that winter velocities tend to be low after high melt summers. presumably this is because it takes longer to shut down the subglacial drainage system after a high melt summer?

We have clarified that this is a consequence of subglacial drainage systems (lines 195-197).

176: role for forcing by.....

183: predict the future mass balance of the Antarctic Ice Sheet and its contribution to sea level change.

318: swaths

The above minor amendments have been corrected.

Martin Sharp, University of Alberta